# Generation and Evaluation of Synthetic Computed Tomography (CT) from Cone-Beam CT (CBCT) by Incorporating Feature-Driven Loss into Intensity-Based Loss Functions in Deep Convolutional Neural Network

**DOI:** 10.3390/cancers14184534

**Published:** 2022-09-19

**Authors:** Sang Kyun Yoo, Hojin Kim, Byoung Su Choi, Inkyung Park, Jin Sung Kim

**Affiliations:** 1Department of Radiation Oncology, Yonsei Cancer Center, Heavy Ion Therapy Research Institute, Yonsei University College of Medicine, Seoul 03722, Korea; 2Medical Physics and Biomedical Engineering Lab. (MPBEL), Yonsei University College of Medicine, Seoul 03722, Korea; 3Medical Artificial Intelligence and Automation (MAIA) Laboratory, Department of Radiation Oncology, University of Texas Southwestern Medical Center, Dallas, TX 75390, USA; 4Oncosoft Inc., Seoul 03787, Korea

**Keywords:** cone-beam computed tomography (CBCT), synthetic computed tomography (CT), convolutional neural network (CNN), SSIM loss, perceptual loss, feature mapping ratio (FMR)

## Abstract

**Simple Summary:**

Despite numerous benefits of cone-beam computed tomography (CBCT), its applications to radiotherapy were limited mainly due to degraded image quality. Recently, enhancing the CBCT image quality by generating synthetic CT image by deep convolutional neural network (CNN) has become frequent. Most of the previous works, however, generated synthetic CT with simple, classical intensity-driven loss in network training, while not specifying a full-package of verifications. This work trained the network by combining feature- and intensity-driven losses and attempted to demonstrate clinical relevance of the synthetic CT images by assessing both image similarity and dose calculating accuracy throughout a commercial Monte-Carlo.

**Abstract:**

Deep convolutional neural network (CNN) helped enhance image quality of cone-beam computed tomography (CBCT) by generating synthetic CT. Most of the previous works, however, trained network by intensity-based loss functions, possibly undermining to promote image feature similarity. The verifications were not sufficient to demonstrate clinical applicability, either. This work investigated the effect of variable loss functions combining feature- and intensity-driven losses in synthetic CT generation, followed by strengthening the verification of generated images in both image similarity and dosimetry accuracy. The proposed strategy highlighted the feature-driven quantification in (1) training the network by perceptual loss, besides L1 and structural similarity (SSIM) losses regarding anatomical similarity, and (2) evaluating image similarity by feature mapping ratio (FMR), besides conventional metrics. In addition, the synthetic CT images were assessed in terms of dose calculating accuracy by a commercial Monte-Carlo algorithm. The network was trained with 50 paired CBCT-CT scans acquired at the same CT simulator and treatment unit to constrain environmental factors any other than loss functions. For 10 independent cases, incorporating perceptual loss into L1 and SSIM losses outperformed the other combinations, which enhanced FMR of image similarity by 10%, and the dose calculating accuracy by 1–2% of gamma passing rate in 1%/1mm criterion.

## 1. Introduction

Image-guided radiotherapy (IGRT) is a viable option in modern radiotherapy [1,2]. Such imaging systems as in-room megavoltage CT, CT-on-rails, and cone-beam CT (CBCT) are available for IGRT [3]. Of those, CBCT equipped with a flat-panel kV detector and kV radiation source became a dominant scanning system over the past decades [4,5,6]. The strength of the imaging system is to provide a 3D volumetric information between patient set-up and actual treatment with relatively low radiation exposure than the other options, followed by facilitating the image matching against the planning CT image.

Over the treatment, the anatomic change derived from a combination of treatment response, weight loss, and radiation effects on normal tissues is inevitable. The changes in internal organs and surface must be non-trivial especially in intensity modulated radiotherapy (IMRT), currently dominant radiation treatment scheme, that try to maximally conform the dose to the target volume, and avoid organ-at-risks (OARs) than the 3D conformal radiotherapy (3D-CRT) [7,8,9]. The notion of on-line and off-line adaptive radiation therapy (ART) [10] was introduced to take preemptive option on the worst case scenario. Volumetric image scanning on a daily fractional basis is also able to monitor the internal anatomic changes, and to help conduct ART in many aspects.

Despite many advantages and potentials above, the limitations of CBCT converge to the image quality. In contrast with conventional fan-beam CT scanners, the cone-shape image scanning increases the unwanted, unexpected scatters reaching to the flat-panel, finally leading to several types of imaging artifacts [11,12]. To overcome the shortcomings in image quality of CBCT, various techniques have been developed. Histogram matching (HM) method [13,14,15] was proposed to enhance dose calculating accuracy. However, matching the HU values of the CT, comprehending the cone-beam artifacts, to those of the planning CT could be problematic. The Monte Carlo (MC) simulation study [16,17,18] and iterative reconstruction method [19,20,21] showed enhancing CBCT image quality for ART at the substantial cost of computational expenses. Although the computation time was significantly reduced with modified MC simulation [22], the additional processing was still demanded to reduce the remaining cone-beam artifacts.

The advent of deep learning based on convolutional neural network (CNN) opened a new prospect in medical image applications, including image segmentation, reconstruction and translation. The image translation, especially, can be applied to enhancing the image quality of CBCT by generating the planning CT-like synthetic image. As most of the planning and dose calculating tasks for radiotherapy are conducted through planning CT images, generating the synthetic CT from CBCT is the rational trial to attempt. Previously, several studies [23,24] developed U-Net based model for generating synthetic CT (sCT) from CBCT with a comparison of image similarity only. Another studies implemented CycleGAN model for the sCT generation providing a promising results [25,26,27,28,29], while CycleGAN may not be an optimal option to take when the paired datasets are available [30].

The previous, recent works, briefly, paid attention to plugging the new network architectures to enhance the performance of reducing the cone-beam artifacts in synthetic CT generation. It did not provide radiation therapy applicable assessment, such as dose calculating accuracy beyond image similarity of the generated synthetic CT to the ground-truth CTs, which may not be sufficient to attain clinical applicability regarding radiotherapy (RT). Many of the studies were also oriented to training the network by intensity-based loss functions accustomed in image translations, possibly undermining the importance of promoting similarity in image features [31,32]. Thus, this study attempted to make contributions to improving the performance in synthetic CT generation as follows:

Incorporating the feature-driven operations into
defining the loss function in training the network for synthetic CT generationassessing the image similarity for the generated image

Offering a full package of verifications for the generated synthetic CT images from CBCT with respect to image similarity throughout dosimetry accuracy

More specifically, this work combined a feature-driven loss function [33,34] with the intensity-based losses, such that it can promote the image feature similarity of the generated images in addition to the anatomical similarity (Section 2.1, Section 2.2 and Section 2.3), as implemented in some of the previous works for the image reconstruction and MR-to-CT image translation [35,36]. In evaluating image similarity, besides the conventional metrics, it employed a feature-oriented metrics to quantify the degree of similarity in image feature (Section 2.4). In an effort to provide a full verifying procedure, the dosimetry accuracy was also investigated for the synthetic CT images by dose calculation throughout a commercial Monte-Carlo algorithm (Section 2.4). Overall, this study was to explore which combination of loss functions for network training would yield the clinically feasible synthetic CT images from CBCT images for RT. To achieve so, it also focused on acquiring well-paired, consistent CT and CBCT images by deformable image registration and by obtaining a pair of CT-CBCT images from a single simulator and treatment unit.

## 2. Materials and Methods

### 2.1. Dataset

The patient cohort for DL model consisted of 65 brain and HN cancer patients with CBCT and CT images, where the patients were scanned in Canon Aquilion LB CT simulator (Canon Medical Systems Corporation, Otawara, Japan), and treated by Elekta Infinity (Elekta, Stockholm, Sweden) linear accelerators from January in 2019 to December in 2021. 52 cases of those were the patients with skull and brain cancers, while the remaining cases were the head-and-neck cancers. Table 1 specifies the characteristics of data used in this work. The 65 scans from 65 patients were divided into 50, 5 and 10 for training, validating, and testing the network, respectively. The input and output images of the network were the paired images of CBCT and planning CT images, which plays a role in generating synthetic CT from CBCT images. This work took a special care of pairing CT and CBCT images and maintaining the data consistency. Namely, to achieve so, we acquired the CT and CBCT images that were scanned and treated at the same imaging simulator and the same treatment unit. Thus, it adopted a general deep neural network, in contrast with cycling GAN architecture suited for unpaired dataset. Hence, he planning CT was deformably registered and resampled to CBCT, named deformed CT (dCT), such that it can enhance the image structural similarity between CBCT and planning CT images.

### 2.2. Loss Functions on FC DensNet

In generating the synthetic CT from CBCT, the training model used a couple of combinations of different loss functions: L1-loss, perceptual loss, and structural similarity (SSIM) loss, as illustrated in Figure 1A. The loss functions used in this work could be split into two types, where the L1 and SSIM loss referred to the anatomical difference in enhancing image similarity. Contrarily, the perceptual loss penalizes and updates the weights in network by comparing features on the images.

The L1-loss function, most frequently used function in DL models with images, measures the pixel-wise mean absolute difference between the output of the DL model and ground-truth. The L1 Loss is expressed in (1):(1)$$L_{L1}\left(x,G\left(x\right)\right)={\left|\left|x-G\left(x\right)\right|\right|}_{1}$$
where x is the true planning CT image, and G(x) is the synthetic CT image generated from CBCT.SSIM has been widely employed as one of the metrics to evaluate image quality. It has the characteristic of preserving image contrast and luminance better than other losses. The SSIM loss is defined as in the following:(2)$$L_{SSIM}\left(x,G\left(x\right)\right)=\frac{\left(2\mu_{x}\mu_{G\left(x\right)}+c_{1}\right)\left(2\sigma_{xG\left(x\right)}+c_{2}\right)}{\left(\mu_{x}^{2}+\mu_{G\left(x\right)}^{2}+c_{1}\right)\left(\sigma_{x}^{2}+\sigma_{G\left(x\right)}^{2}+c_{2}\right)}$$
where μ𝑥 is the average of x, μ𝐺(𝑥) is the average of G(x), σ𝑥 2 is the variance of x, σ𝐺(𝑥) 2 is the variance of G(x), and 𝜎𝑥𝐺(𝑥) is the covariance of x and G(x), 𝑐 is the stabilization variable. The perceptual loss function was first introduced as a VGG loss based on a pre-trained VGG network. The perceptual loss compensates for the perceptually unsatisfactory results of pixel-wise losses such as L1 Loss. For this, perceptual loss calculates the euclidean distance between feature maps extracted from a pre-trained VGG network. The definition of perceptual loss is given in (3):(3)$$L_{Perceptual}\left(x,G\left(x\right)\right)={\left|\left|VGG\left(x\right)-VGG\left(G\left(x\right)\right)\right|\right|}_{2}^{2}$$

The loss functions defined above were plugged into one of the fully convolutional network (FCN) structures, FC DenseNet [37]. The DenseNet was employed in this application as it t has shown a promising result for the paired image dataset. FCN is similar to common CNN, while the fully connected layers are removed from the end of the network and the output is generated by combining the output of pooling layers from different convolutional ones. As shown in Figure 1B, the FC-DenseNet consisted of down-sampling path, up-sampling path. Specifically, in down-sampling path, following the convolution layer, the transition down layers consists of batch normalization, exponential linear units (ELU) [38], 1 × 1 convolution, dropout (*p* = 0.2), and a 2 × 2 max-pooling operation. In up-sampling path, the transition up layers, before the convolution layer, consist of transposed convolution.

### 2.3. Implementation

The deformable image registration (DIR) between CBCT (target) and planning CT (moving) images was performed on MIM software (Mim Software Inc., Cleveland, OH, USA) and MATLAB (The MathWorks, Inc., Natick, MA, USA) with deformation and resampling, the resolution and imaging size of deformed planning CT image (dCT) are identical to those of CBCT (270 × 270 with 1 mm resolution). The FC-DenseNet was implemented using Python 3.8.3 and PyTorch 1.5.1 [39]. The training was performed on graphical processing units (GPUs) (NVIDIA TITAN RTX with 24 GB of memory). Each model was trained under the identical hyper-parameter setting, while varying the definition of loss functions. The number of epochs was 150. The DL architecture had an initial learning rate of 0.00002 and the architecture’s weights were optimized using Adam [40]. During the training, data augmentations (horizontal flip, random rotation, random blur) were applied on the fly randomly.

From our observations, the image similarity was enhanced when the perceptual and/or SSIM losses were combined with L1-loss. Thus, we composed the loss function by addition of L1-loss to perceptual loss and/or SSIM loss. The combinations exploited in this work were defined as follows: L1-loss only (L1), L1 + Perceptual loss (LP), L1 + SSIM loss (LS), and L1 + Perceptual + SSIM loss (LPS). In this work, the weights for the different losses were well-balanced as imposing unbalanced regularizing weights led to poorer performance from our observations.

### 2.4. Evaluation

#### 2.4.1. Image Similarity

To evaluate the image similarity, the dCT images were used as the ground truth to evaluate the four different loss combination models. We compared the image similarity between ground-truth dCT and synthetic CT images in terms of such conventional similarity metrics as mean-absolute error (MAE), SSIM, and peak-signal-to-noise-ratio (PSNR).

Recently, interest in features of tomographic images such as CT, magnetic resonance (MR), or positron emission tomography (PET) images has increased [41]. There are SIFT, KAZE, ORB, etc. methods for mapping features from images. Although the above image similarity metrics are good evaluations, one step further, we propose a feature mapping ratio (FMR) based on the A-KAZE feature mapping algorithm [42], an algorithm that improved the SIFT algorithm, as an image similarity comparison metric. Figure 2 shows how FMR processes the feature points in the paired sCT and dCT images. It extracts feature points in each sCT and dCT images, following calculates binary descriptors, and matches descriptors. Next, the image quality was compared through the ratio between the detected features and the matched features. Specifically, in the step of extracting feature points, it computes a nonlinear scale space to extract feature points, generating a robust binary descriptor that exploits gradient information from the nonlinear scale space. In the following descriptor matching, a brute-force algorithm is used to match descriptors with a hamming distance less than a threshold, which was set to be 0.8 in this work.

#### 2.4.2. Dose Calculation

We had an optimized VMAT plan for each testing case, designed to treat the patient on the given CT image. As the deformed CT images (dCT) were slightly different in registration, the same planning parameters were applied to the dCTs. Hence, the calculated dose on dCT was defined as the ground-truth dose distribution. The dose calculation was performed by MONACO treatment planning system from Elekta, in which the commercial Monte-Carlo (MC) dose calculation is available. The dose distributions on the synthetic CT images resulted from the different loss functions were calculated by the commercial MC algorithm with the same VMAT planning parameters as those for the dCT images. The dosimetric similarity was quantified by absolute dose difference, and gamma passing rate on 1%/1 mm, and 2%/2 mm criteria.

## 3. Results

### 3.1. Image Similarity Comparison

Figure 3 illustrates a couple of exemplary synthetic CT images of the 10 testing cases, generated from different loss functions. It shows that the synthetic CT images produced from LS and LPS loss functions tended to be analogous to the ground-truth CT images. As indicated by arrows, the synthetic CT images from the SSIM-associated loss functions well simulated the structural details of the true CT images, relative to those from L1-loss and LP loss functions.

Table 2 lists the numerical details of image similarity between the dCT and sCT images for the 10 testing cases. The synthetic CT images generated from different loss functions resulted in about 6 HU distance on average against the ground-truth CT images, which were fairly good. In SSIM, MAE, and PSNR, it turns out that the LPS loss produced slightly more accurate synthetic CT images than the other loss functions though the differences were not significant. In feature matching throughout the FMR, the difference across different loss function definitions became more explicit. Combining the SSIM loss with L1-loss increased the FMR from 0.532 to 0.569. With all loss function (L1, SSIM, and perceptual losses) combined, FMR reached out to 0.683. The perceptual loss resulted in the worse image similarity in the conventional metrics, even relative to L1-loss only. It is interesting to note that the LP loss produced higher value than L1-loss on average in feature matching. It could be associated with the fact that the perceptual loss was designed to put more emphasis on the feature representation.

### 3.2. Dose Distribution Comparison

Table 3 shows the dosimetric analysis with gamma passing rate and absolute dose difference across the 10 cases. It compared the reference and compute dose distributions calculated on dCT and sCTs from different loss functions. Though LP loss function produced greater gamma passing rate than L1-loss in 2%/2 mm criterion of gamma passing rate, LPS was followed by LP, L1 and LS in dosimetric accuracy. It was obvious that the dose distribution computed on the synthetic CT images from LPS had the lowest error to the ground-truth dose distribution with no exception. In 1%/1 mm criterion of gamma passing rate, the LPS loss function had 96.2% passing rate on average, which was about 1% and 1.5% greater than those from the L1 and LP loss functions.

Figure 4 illustrates the dosimetric comparison between dose distributions on dCT and sCTs. From absolute dose differences in the second row of Figure 4, the magnitude of dosimetric errors got smaller from left to right (from L1-loss to LPS loss functions). In addition, in the third row of Figure 4 representing gamma passing rate, it was seen that the region of the errors became narrower on the synthetic CT images from SSIM-associated loss functions.

## 4. Discussion

CBCT has many desirable features suitable for IGRT, whereas the degraded image quality relative to the planning CT images were pitfalls. The source of the degeneracy was mainly caused by cone-shape imaging with flat-panel detector, which was inherently susceptible to the photon scattering. The so-called cone-beam artifact reduction was made easier with an aid of an emerging framework, CNNs with deep learning. Many studies attempted to enhance the CBCT image quality by generating the synthetic CT images from CBCT images throughout deep convolutional neural networks. It was found that most of the studies applied the new network architectures to synthetic CT generation, while a few studies provided a full package including image similarity and dosimetric analysis to see if the algorithm is applicable to the clinic. In addition, most of the works utilized the intensity-based loss function in training the given network, overlooking the possibility of promoting similarity in image features.

To differentiate from the previous studies, this work derived the feature-driven quantifications in defining loss function by perceptual loss and evaluating image similarity by the means of feature mapping ratio (FMR). In training the network, in addition to the perceptual loss, we diversified the loss function with SSIM loss combined with L1-loss, which could also strengthen the anatomical and structural similarities. In assessing the dosimetric accuracy, the dose calculation was performed on the MC-based algorithm with actual VMAT planning parameters. To make it fully controlled, the other environmental variables were constrained, such that all CBCT and CT images used for training and testing the network were obtained from the only one RT treatment unit and the same CT simulator.

As seen in the results, the SSIM-associated loss function (L1 + SSIM, and L1 + perceptual + SSIM) produced the most similar synthetic CT images to the ground-truth CT images with respect to the image-similarity metrics. The perceptual loss did not lead to better results than L1-loss only in conventional similarity metrics. Contrarily, the synthetic CT images from LP (L1 + perceptual) loss were greater than those from L1-loss only on average. This implies that the feature driven training affected the feature-mapping accuracy constructively. The feature-based perceptual loss was demonstrated to be powerful if it is combined with secure structural similarity, in which the LPS loss (L1 + perceptual + SSIM)-based training enlarged the FMR significantly without compromising the structural similarity as carried out by the LP loss. The similar tendency was appeared in analyzing dosimetric accuracy. The synthetic CT images generated by the LPS loss outperformed the other images from different combinations in both gamma passing rate and absolute dose differences. The synthetic CT images generated from CBCT resulted in 96.2% and 99.6% gamma passing rates against the reference dose distribution in 1%/1 mm, and 2%/2 mm.

The low-grade image quality of CBCT has been able to affect the accuracy of IGRT. The reduction in cone-beam artifacts directly facilitates the image matching procedure in IGRT. Especially, the abovementioned results regarding dosimetric accuracy would be able to extend the border of CBCT. To be more specific, by generating synthetic CT from CBCT, the dose calculation on the synthetic CT would become available, which are potentially used for evaluating dose distribution and summation on a fraction basis. In addition, it could reduce the necessity of CT re-simulation by possible substitute CT images for synthetic CT images from CBCT. The synthetic CT images having greater image contrast than CBCT with reduced imaging artifacts would be beneficial for the deep learning-based auto-segmentation as well. Hence, these factors mostly associated with enhanced efficiency could eventually facilitate the realization of adaptive radiation therapy (ART).

Despite various advantages delivered from this work, there are a couple of limitations to be stated. To stand out the benefits of this work, we constrained the variability of CT/CBCT image data. From our perspectives, however, it was more important to examine which combination of loss functions can produce more qualified results with a constrained dataset. Another potential limitation might have been increase in inefficiency due to additional loss functions in the network training. In fact, the training time was increased by about 60% with additional 0.6 GB GPU VRAM usage when accompanying the perceptual loss that requires for running the VGG network. With respect to the inference after completing the training, which is considered more important in clinical aspects, there was no such a big difference in time, presenting only 1 s (5 s vs. 6 s). The body sites that we referred to were brain and head-and-neck regions, which were covered by CBCT images. In upper abdomen and pelvic regions, the maximum FOV of CBCT may not cover whole region of interest. In such conditions, the CBCT images were further influenced by the additional scattering effect adjacent to the marginal side of images. Last, the synthetic CT images from CBCT were to reduce so-called cone-beam artifacts, while it possibly has an inherent CT-possessing artifacts. There is room, thus, to further enhance the image quality and CBCT-based (adaptive) radiotherapy.

## 5. Conclusions

This study investigated the generation of synthetic CT from CBCT to reduce the cone-beam artifacts, thus enhancing the image quality of CBCT. We varied the definition of loss functions combining L1-loss with intensity and shape-based SSIM, and feature-based perceptual losses for the well registered, paired CBCT-CT dataset. With evaluating metrics including image similarity by feature mapping criterion, and dosimetric accuracy for the MC-simulated dose distributions, the SSIM-associated loss functions produced the qualified synthetic CT images. When incorporating the perceptual loss into L1- and SSIM losses, the resulting synthetic CT images yielded the best performance in both image similarity and dose calculating accuracy. The results would support a claim that CBCT, once being developed to reduce the artifacts, could be employed for radiation therapy in more constructive ways, such as for adaptive radiation therapy.

## Figures and Tables

**Figure 1 cancers-14-04534-f001:**
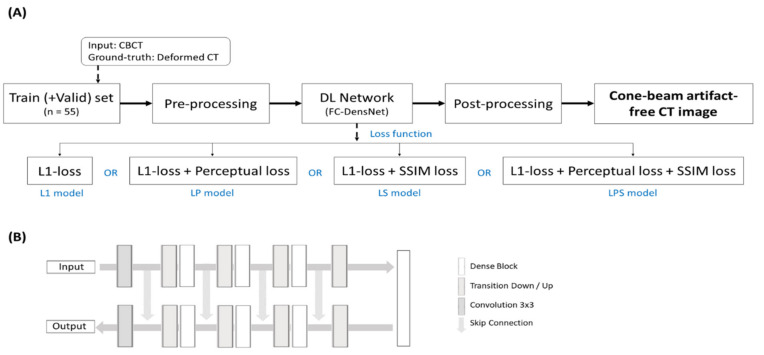
Study design of our study. (**A**) Flowchart of the compare four different learning losses and obtain a cone-beam artifact-free CT image trained with each loss. (**B**) FC-DenseNet is used as the DL architecture for the eliminate cone-beam artifact in CBCT.

**Figure 2 cancers-14-04534-f002:**
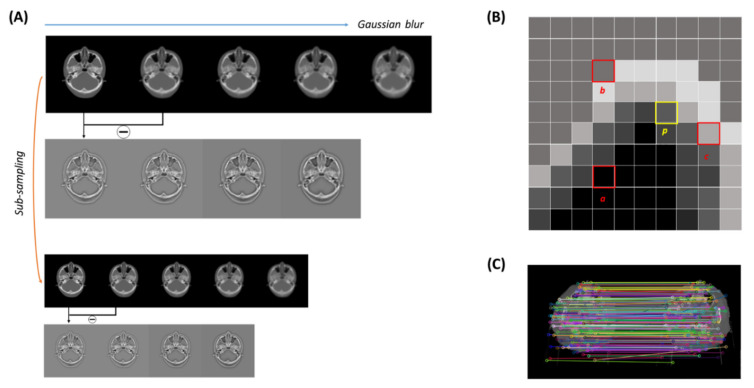
Feature mapping ratio (FMR). (**A**) Example of detect feature point. To detect the feature point, the difference image between adjacent Gaussian blurring images is used and the local extrema position is used as the feature point. (**B**) Example of binary descriptor. The binary number is calculated as 110(2) by points a, b, and c around the feature point p. 110(2) means information that b is brighter than a, c is brighter than b, and a is darker than c. (**C**) The binary descriptor determines the similarity using a Hamming distance.

**Figure 3 cancers-14-04534-f003:**
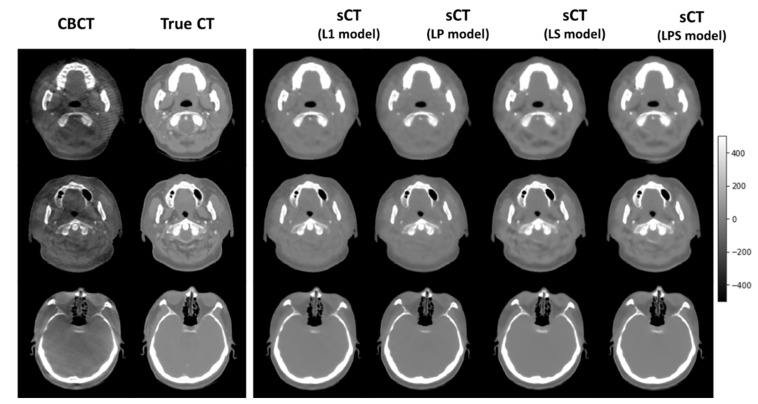
Examples of DL output images. True CT, CBCT, synthetic CT(L1 loss), synthetic CT(LP loss), synthetic CT(LS loss), and synthetic CT(LPS loss) from axial plane from one of the patients in the 10 test set.

**Figure 4 cancers-14-04534-f004:**
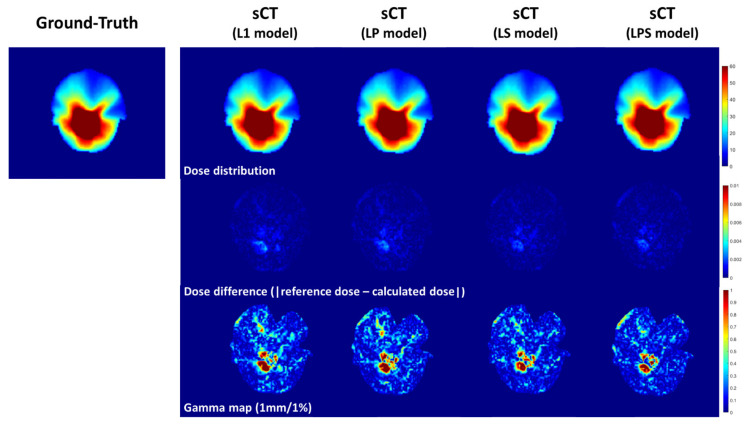
Example of dose distributions and dose differences. Ground-Truth, synthetic CT(L1 loss), synthetic CT(LP loss), synthetic CT(LS loss), and synthetic CT(LPS loss) from axial plane from Patient 3 in the 10 test set.

**Table 1 cancers-14-04534-t001:** Patient Characteristics.

Variables	Total (65)	Train (+Valid) Set (55)	Test Set (10)
**Age (years)**			
Median (range)	56 (3–83)	57.5 (3–78)	55.5 (24–83)
**Sex**			
Male	33 (51)	27 (49)	6 (60)
Female	32 (49)	28 (51)	4 (40)
**Acquisition Diff. (days)**			
Median (range)	13 (0–119)	13 (0–119)	15.5 (7–33)

Abbreviations: Acquisition Diff., acquisition date difference between CBCT images and CT images.

**Table 2 cancers-14-04534-t002:** SSIM, MAE (HU), and FMR performances for the DL outputs by generated from different learning losses for the 10 test set.

		P1	P2	P3	P4	P5	P6	P7	P8	P9	P10	Average
SSIM	L1	0.995	0.980	0.988	0.986	0.980	0.986	0.988	0.991	0.985	0.991	0.987 (±0.045)
	LP	0.993	0.979	0.988	0.984	0.978	0.984	0.987	0.989	0.984	0.990	0.986 (±0.045)
	LS	0.993	0.980	0.988	0.986	0.983	0.985	0.989	0.990	0.985	0.988	0.987 (±0.035)
	LPS	0.995	0.983	0.991	0.987	0.985	0.988	0.990	0.992	0.987	0.992	**0.989 (±0.034)**
MAE (HU)	L1	3.324	8.999	5.101	7.345	9.550	6.562	6.555	4.618	7.790	4.546	6.439 (±1.931)
	LP	3.755	9.049	4.985	7.644	9.888	6.825	6.658	4.831	8.349	4.828	6.681 (±1.947)
	LS	3.513	8.662	5.015	7.340	8.754	6.629	6.199	4.807	7.750	5.192	6.386 (±1.664)
	LPS	3.211	7.954	4.385	6.541	8.016	6.080	5.724	4.289	7.309	4.421	**5.793 (±1.592)**
PSNR	L1	41.862	33.787	37.982	36.368	34.067	35.817	36.403	39.431	36.383	38.652	37.075 (±2.338)
	LP	41.406	34.048	38.568	36.249	33.899	35.734	36.793	39.506	36.356	38.616	37.118 (±2.271)
	LS	41.874	34.652	38.435	36.650	35.202	35.905	37.374	39.340	36.879	37.590	37.390 (±2.008)
	LPS	42.418	34.991	39.466	31.173	35.553	36.155	37.469	40.024	36.973	38.662	**37.888 (±2.157)**
FMR(0.8)	L1	0.605	0.484	0.601	0.508	0.444	0.490	0.459	0.622	0.526	0.585	0.532 (±0.062)
	LP	0.613	0.470	0.623	0.525	0.433	0.502	0.468	0.605	0.549	0.601	0.539 (±0.066)
	LS	0.642	0.503	0.629	0.574	0.510	0.530	0.513	0.613	0.580	0.592	0.569 (±0.049)
	LPS	0.644	0.530	0.646	0.578	0.534	0.535	0.536	0.656	0.562	0.604	**0.683 (±0.049)**

**Table 3 cancers-14-04534-t003:** Gamma passing rate and dose difference analysis on CBCT and DL output by different learning losses compared with dose distribution on deformed CT for the 10 test set.

			P1	P2	P3	P4	P5	P6	P7	P8	P9	P10	Average
**Gamma passing rate**	1 mm/1%	L1	0.9885	0.9101	0.9909	0.9731	0.9727	0.8988	0.9094	0.9707	0.9027	0.9388	0.9456 (±0.036)
		LP	0.9869	0.8911	0.9853	0.9763	0.9685	0.8913	0.8851	0.9650	0.9840	0.9267	0.9370 (±0.041)
		LS	0.9836	0.9110	0.9856	0.9819	0.9821	0.9285	0.9190	0.9714	0.9123	0.9338	0.9510 (±0.031)
		LPS	0.9919	0.9279	0.9919	0.9894	0.9909	0.9390	0.9472	0.9801	0.9225	0.9374	**0.9618 (±0.028)**
	2 mm/2%	L1	0.9999	0.9869	0.9999	0.9954	0.9954	0.9851	0.9884	0.9982	0.9865	0.9933	0.9929 (±0.005)
		LP	0.9999	0.9849	0.9997	0.9980	0.9935	0.9850	0.9886	0.9980	0.9904	0.9953	0.9933 (±0.006)
		LS	0.9999	0.9919	0.9998	0.9993	0.9986	0.9911	0.9943	0.9978	0.9926	0.9933	0.9960 (±0.003)
		LPS	1.0000	0.9947	0.9999	0.9997	0.9998	0.9938	0.9968	0.9992	0.9885	0.9969	**0.9969 (±0.004)**
Dose difference		L1	0.0132	0.0146	0.0063	0.0081	0.0074	0.0197	0.0174	0.0100	0.0163	0.0125	0.0126 (±0.004)
		LP	0.0134	0.0155	0.0068	0.0080	0.0083	0.0207	0.0180	0.0104	0.0175	0.0129	0.0132 (±0.005)
		LS	0.0142	0.0145	0.0068	0.0077	0.0083	0.0174	0.0161	0.0102	0.0149	0.0134	0.0124 (±0.004)
		LPS	0.0125	0.0129	0.0060	0.0066	0.0081	0.0150	0.0141	0.0095	0.0151	0.0126	**0.0112 (±0.003)**

## Data Availability

The original contributions presented in the study are included in the article. Further inquiries can be directed to the corresponding author.

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
