# Peer review of "Generation and Evaluation of Synthetic Computed Tomography (CT) from Cone-Beam CT (CBCT) by Incorporating Feature-Driven Loss into Intensity-Based Loss Functions in Deep Convolutional Neural Network"

_cancers, 2022, doi:10.3390/cancers14184534_

Round 1

Reviewer 1 Report

The paper is very interesting and very well written. The following minor changes should be considered. 

1. Expand the introduction if possible, and at the end of the Introdcution explicitly state hypothesis of your research, preferably in bullet form. Also add another paragraph describing the following sections of your paper.

Reviewer 2 Report

The merit of the proposed approach is supported by the results, but I miss on the paper a bit more discussion on why these techniques were chosen for this problem and had not been considered before. This however is more of a nitpicking than a detrimental comment.

The introduction is not clear and very less literature is used. Follow these instruction: The introduction should briefly place the study in a broad context and highlight why it is important. It should define the purpose of the work and its significance, including specific hypotheses being tested. The current state of the research field should be reviewed carefully and key publications cited. Please highlight controversial and diverging hypotheses when necessary. Finally, briefly mention the main aim of the work and highlight the main conclusions. Keep the introduction comprehensible to scientists working outside the topic of the paper.

Below papers has some interesting implications that you could discuss in your introduction and how it relates to your work.

Vulli, A.; et al.. Fine-Tuned DenseNet-169 for Breast Cancer Metastasis Prediction Using FastAI and 1-Cycle Policy. Sensors 2022, 22, 2988.

Singh, Jaiteg, et al. "Classification and analysis of android malware images using feature fusion technique." IEEE Access 9 (2021): 90102-90117.

What was the key motivation behind focusing on the Deep Learning?

It would be interesting if the authors report the trade-off compared to other methods especially the computational complexity of the models. Some techniques require more memory space and take longer time, please elaborate on that. 

Another dataset would consolidate the work if the authors obtain a consisting result. 

Authors should further clarify and elaborate novelty in their contribution.

What are the limitations of the present work?

What are the practical implications of this research?

Round 2

Reviewer 2 Report

.